# A RECURRENT NEURAL NETWORK WITHOUT CHAOS

**Thomas Laurent**
Department of Mathematics
Loyola Marymount University
Los Angeles, CA 90045, USA
tlaurent@lmu.edu

**James von Brecht**
Department of Mathematics
California State University, Long Beach
Long Beach, CA 90840, USA
james.vonbrecht@csulb.edu

## ABSTRACT

We introduce an exceptionally simple gated recurrent neural network (RNN) that achieves performance comparable to well-known gated architectures, such as LSTMs and GRUs, on the word-level language modeling task. We prove that our model has simple, predicable and non-chaotic dynamics. This stands in stark contrast to more standard gated architectures, whose underlying dynamical systems exhibit chaotic behavior.

## 1 INTRODUCTION

Gated recurrent neural networks, such as the Long Short Term Memory network (LSTM) introduced by Hochreiter & Schmidhuber (1997) and the Gated Recurrent Unit (GRU) proposed by Cho et al. (2014), prove highly effective for machine learning tasks that involve sequential data. We propose an exceptionally simple variant of these gated architectures. The basic model takes the form

$$h_t = \theta_t \odot \tanh(h_{t-1}) + \eta_t \odot \tanh(W x_t), \tag{1}$$

where $\odot$ stands for the Hadamard product. The horizontal/forget gate (i.e. $\theta_t$) and the vertical/input gate (i.e. $\eta_t$) take the usual form used in most gated RNN architectures. Specifically

$$\theta_t := \sigma \left( U_\theta h_{t-1} + V_\theta x_t + b_\theta \right) \quad \text{and} \quad \eta_t := \sigma \left( U_\eta h_{t-1} + V_\eta x_t + b_\eta \right) \tag{2}$$

where $\sigma(x) := (1 + \mathrm{e}^{-x})^{-1}$ denotes the logistic sigmoid function. The network (1)–(2) has quite intuitive dynamics. Suppose the data $x_t$ present the model with a sequence

$$(W x_t)(i) = \begin{cases} 10 & \text{if } t = T \\ 0 & \text{otherwise,} \end{cases} \tag{3}$$

where $(W x_t)(i)$ stands for the $i^{\text{th}}$ component of the vector $W x_t$. In other words we consider an input sequence $x_t$ for which the learned $i^{\text{th}}$ feature $(W x_t)(i)$ remains off except at time $T$. When initialized from $h_0 = 0$, the corresponding response of the network to this "impulse" in the $i^{\text{th}}$ feature is

$$h_t(i) \approx \begin{cases} 0 & \text{if } t < T \\ \eta_T & \text{if } t = T \\ \alpha_t & \text{if } t > T \end{cases} \tag{4}$$

with $\alpha_t$ a sequence that relaxes toward zero. The forget gate $\theta_t$ control the rate of this relaxation. Thus $h_t(i)$ activates when presented with a strong $i^{\text{th}}$ feature, and then relaxes toward zero until the data present the network once again with strong $i^{\text{th}}$ feature. Overall this leads to a dynamically simple model, in which the activation patterns in the hidden states of the network have a clear cause and predictable subsequent behavior.

Dynamics of this sort do not occur in other RNN models. Instead, the three most popular recurrent neural network architectures, namely the vanilla RNN, the LSTM and the GRU, have complex, irregular, and unpredictable dynamics. Even in the absence of input data, these networks can give rise to chaotic dynamical systems. In other words, when presented with null input data the activation patterns in their hidden states do not necessarily follow a predictable path. The proposed network (1)–(2) has rather dull and minimalist dynamics in comparison; its only attractor is the zero state,

and so it stands at the polar-opposite end of the spectrum from chaotic systems. Perhaps surprisingly, at least in the light of this comparison, the proposed network (1) performs as well as LSTMs and GRUs on the word level language modeling task. We therefore conclude that the ability of an RNN to form chaotic temporal dynamics, in the sense we describe in Section 2, cannot explain its success on word-level language modeling tasks.

In the next section, we review the phenomenon of chaos in RNNs via both synthetic examples and trained models. We also prove a precise, quantified description of the dynamical picture (3)–(4) for the proposed network. In particular, we show that the dynamical system induced by the proposed network is never chaotic, and for this reason we refer to it as a Chaos-Free Network (CFN). The final section provides a series of experiments that demonstrate that CFN achieve results comparable to LSTM on the word-level language modeling task. All together, these observations show that an architecture as simple as (1)–(2) can achieve performance comparable to the more dynamically complex LSTM.

## 2    Chaos in Recurrent Neural Networks

The study of RNNs from a dynamical systems point-of-view has brought fruitful insights into generic features of RNNs (Sussillo & Barak, 2013; Pascanu et al., 2013). We shall pursue a brief investigation of CFN, LSTM and GRU networks using this formalism, as it allows us to identify key distinctions between them. Recall that for a given mapping $\Phi : \mathbb{R}^d \mapsto \mathbb{R}^d$, a given initial time $t_0 \in \mathbb{N}$ and a given initial state $\mathfrak{u}_0 \in \mathbb{R}^d$, a simple repeated iteration of the mapping $\Phi$

$$\mathfrak{u}_{t+1} = \Phi(\mathfrak{u}_t) \quad t > t_0,$$
$$\mathfrak{u}_{t_0} = \mathfrak{u}_0 \qquad t = t_0,$$

defines a *discrete-time dynamical system*. The index $t \in \mathbb{N}$ represents the current time, while the point $\mathfrak{u}_t \in \mathbb{R}^d$ represents the current state of the system. The set of all visited states $\mathcal{O}^+(\mathfrak{u}_0) := \{\mathfrak{u}_{t_0}, \mathfrak{u}_{t_0+1}, \ldots, \mathfrak{u}_{t_0+n}, \ldots\}$ defines the *forward trajectory* or *forward orbit* through $\mathfrak{u}_0$. An *attractor* for the dynamical system is a set that is invariant (any trajectory that starts in the set remains in the set) and that attracts all trajectories that start sufficiently close to it. The attractors of chaotic dynamical systems are often fractal sets, and for this reason they are referred to as *strange attractors*.

Most RNNs generically take the functional form

$$\mathfrak{u}_t = \Psi(\mathfrak{u}_{t-1}, W_1 x_t, W_2 x_t, \ldots, W_k x_t), \tag{5}$$

where $x_t$ denotes the $t^{\text{th}}$ input data point. For example, in the case of the CFN (1)–(2), we have $W_1 = W, W_2 = V_\theta$ and $W_3 = V_\eta$. To gain insight into the underlying design of the architecture of an RNN, it proves usefull to consider how trajectories behave when they are not influenced by any external input. This lead us to consider the dynamical system

$$\mathfrak{u}_t = \Phi(\mathfrak{u}_{t-1}) \qquad \Phi(\mathfrak{u}) := \Psi(\mathfrak{u}, 0, 0, \ldots, 0), \tag{6}$$

which we refer to as the *dynamical system induced* by the recurrent neural network. The time-invariant system (6) is much more tractable than (5), and it offers a mean to investigate the inner working of a given architecture; it separates the influence of input data $x_t$, which can produce essentially any possible response, from the model itself. Studying trajectories that are not influenced by external data will give us an indication on the ability of a given RNN to generate complex and sophisticated trajectories by its own. As we shall see shortly, the dynamical system induced by a CFN has excessively simple and predictable trajectories: all of them converge to the zero state. In other words, its only attractor is the zero state. This is in sharp contrast with the dynamical systems induced by LSTM or GRU, who can exhibit chaotic behaviors and have *strange attractors*.

The learned parameters $W_j$ in (5) describe how data influence the evolution of hidden states at each time step. From a modeling perspective, (6) would occur in the scenario where a trained RNN has learned a weak coupling between a specific data point $x_{t_0}$ and the hidden state at that time, in the sense that the data influence is small and so all $W_j x_{t_0} \approx 0$ nearly vanish. The hidden state then transitions according to $\mathfrak{u}_{t_0} \approx \Psi(\mathfrak{u}_{t_0-1}, 0, 0, \ldots, 0) = \Phi(\mathfrak{u}_{t_0-1})$.

We refer to Bertschinger & Natschläger (2004) for a study of the chaotic behavior of a simplified vanilla RNN with a specific statistical model, namely an i.i.d. Bernoulli process, for the input data as well as a specific statistical model, namely i.i.d. Gaussian, for the weights of the recurrence matrix.

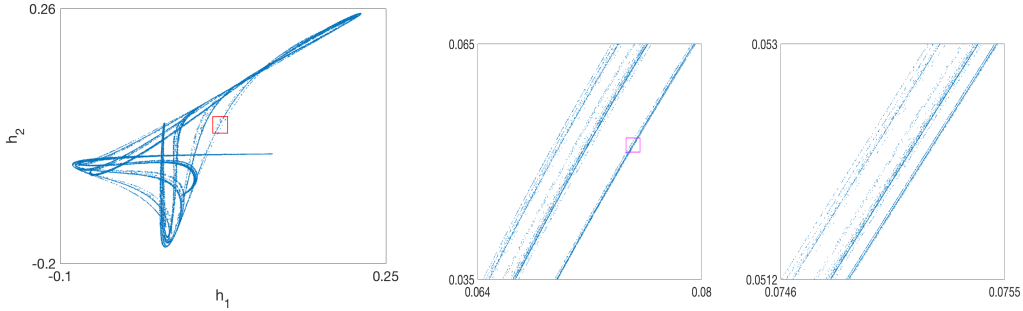

Figure 1: Strange attractor of a 2-unit LSTM. Successive zooms (from left to right) reveal the self-repeating, fractal nature of the attractor. Colored boxes depict zooming regions.

## 2.1 CHAOTIC BEHAVIOR OF LSTM AND GRU IN THE ABSENCE OF INPUT DATA

In this subsection we briefly show that LSTM and GRU, *in the absence of input data*, can lead to dynamical systems $\mathfrak{u}_t = \Phi(\mathfrak{u}_{t-1})$ that are chaotic in the classical sense of the term (Strogatz, 2014). Figure 1 depicts the strange attractor of the dynamical system:

$$\mathfrak{u}_t = \begin{bmatrix} h_t \\ c_t \end{bmatrix} \quad \mathfrak{u} \mapsto \Phi(\mathfrak{u}) = \begin{bmatrix} o \odot \tanh\left(f \odot c + i \odot g\right) \\ f \odot c + i \odot g \end{bmatrix} \tag{7}$$

$$i := \sigma(W_i h + b_i) \quad f := \sigma(W_f h + b_f) \quad o := \sigma(W_o h + b_o) \quad g := \tanh(W_g h + b_g) \tag{8}$$

induced by a two-unit LSTM with weight matrices

$$W_i = \begin{bmatrix} -1 & -4 \\ -3 & -2 \end{bmatrix} \quad W_o = \begin{bmatrix} 4 & 1 \\ -9 & -7 \end{bmatrix} \quad W_f = \begin{bmatrix} -2 & 6 \\ 0 & -6 \end{bmatrix} \quad W_g = \begin{bmatrix} -1 & -6 \\ 6 & -9 \end{bmatrix} \tag{9}$$

and zero bias for the model parameters. These weights were randomly generated from a normal distribution with standard deviation 5 and then rounded to the nearest integer. Figure 1(a) was obtained by choosing an initial state $\mathfrak{u}_0 = (h_0, c_0)$ uniformly at random in $[0,1]^2 \times [0,1]^2$ and plotting the h-component of the iterates $\mathfrak{u}_t = (h_t, c_t)$ for $t$ between $10^3$ and $10^5$ (so this figure should be regarded as a two dimensional projection of a four dimensional attractor, which explain its tangled appearance). Most trajectories starting in $[0,1]^2 \times [0,1]^2$ converge toward the depicted attractor. The resemblance between this attractor and classical strange attractors such as the Hénon attractor is striking (see Figure 5 in the appendix for a depiction of the Hénon attractor). Successive zooms on the branch of the LSTM attractor from Figure 1(a) reveal its fractal nature. Figure 1(b) is an enlargement of the red box in Figure 1(a), and Figure 1(c) is an enlargement of the magenta box in Figure 1(b). We see that the structure repeats itself as we zoom in.

The most practical consequence of chaos is that the long-term behavior of their forward orbits can exhibit a high degree of sensitivity to the initial states $\mathfrak{u}_0$. Figure 2 provides an example of such behavior for the dynamical system (7)–(9). An initial condition $\mathfrak{u}_0$ was drawn uniformly at random in $[0,1]^2 \times [0,1]^2$. We then computed $100,000$ small amplitude perturbations $\hat{\mathfrak{u}}_0$ of $\mathfrak{u}_0$ by adding a small random number drawn uniformly from $[-10^{-7}, 10^{-7}]$ to each component. We then iterated (7)–(9) for 200 steps and plotted the h-component of the final state $\hat{\mathfrak{u}}_{200}$ for each of the $100,000$ trials on Figure 2(a). The collection of these $100,000$ final states essentially fills out the entire attractor, despite the fact that their initial conditions are highly localized (i.e. at distance of no more than $10^{-7}$) around a fixed point. In other words, the time $t = 200$ map of the dynamical system will map a small neighborhood around a fixed initial condition $\mathfrak{u}_0$ to the entire attractor. Figure 2(b) additionally illustrates this sensitivity to initial conditions for points on the attractor itself. We take an initial condition $\mathfrak{u}_0$ on the attractor and perturb it by $10^{-7}$ to a nearby initial condition $\hat{\mathfrak{u}}_0$. We then plot the distance $\|\hat{\mathfrak{u}}_t - \mathfrak{u}_t\|$ between the two corresponding trajectories for the first 200 time steps. After an initial phase of agreement, the trajectories strongly diverge.

The synthetic example (7)–(9) illustrates the potentially chaotic nature of the LSTM architecture. We now show that chaotic behavior occurs for *trained* models as well, and not just for synthetically generated instances. We take the parameter values of an LSTM with 228 hidden units trained on the

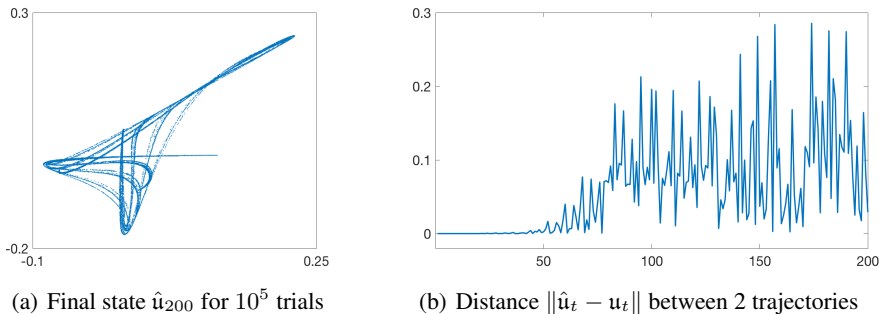

(a) Final state $\hat{u}_{200}$ for $10^5$ trials (b) Distance $\|\hat{u}_t - u_t\|$ between 2 trajectories

Figure 2: (a): A small neighborhood around a fixed initial condition $u_0$, after 200 iterations, is mapped to the entire attractor. (b): Two trajectories starting starting within $10^{-7}$ of one another strongly diverge after 50 steps.

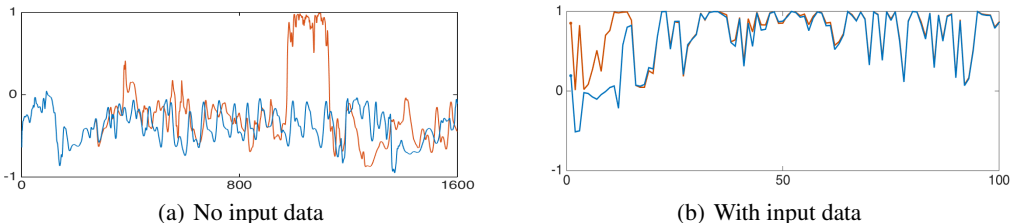

(a) No input data (b) With input data

Figure 3: 228-unit LSTM trained on Penn Treebank. (a): In the absence of input data, the system is chaotic and nearby trajectories diverge. (b): In the presence of input data, the system is mostly driven by the external input. Trajectories starting far apart converge.

Penn Treebank corpus without dropout (c.f. the experimental section for the precise procedure). We then set all data inputs $x_t$ to zero and run the corresponding induced dynamical system. Two trajectories starting from nearby initial conditions $u_0$ and $\hat{u}_0$ were computed (as before $\hat{u}_0$ was obtained by adding to each components of $u_0$ a small random number drawn uniformly from $[-10^{-7}, 10^{-7}]$). Figure 3(a) plots the first component h(1) of the hidden state for both trajectories over the first 1600 time steps. After an initial phase of agreement, the forward trajectories $\mathcal{O}^+(u_0)$ and $\mathcal{O}^+(\hat{u}_0)$ strongly diverge. We also see that both trajectories exhibit the typical aperiodic behavior that characterizes chaotic systems. If the inputs $x_t$ do not vanish, but come from actual word-level data, then the behavior is very different. The LSTM is now no longer an autonomous system whose dynamics are driven by its hidden states, but a time dependent system whose dynamics are mostly driven by the external inputs. Figure 3(b) shows the first component $h(1)$ of the hidden states of two trajectories that start with initial conditions $u_0$ and $\hat{u}_0$ that are far apart. The sensitivity to initial condition disappears, and instead the trajectories converge toward each other after about 70 steps. The memory of this initial difference is lost. Overall these experiments indicate that a trained LSTM, when it is not driven by external inputs, can be chaotic. In the presence of input data, the LSTM becomes a forced system whose dynamics are dominated by external forcing.

Like LSTM networks, GRU can also lead to dynamical systems that are chaotic and they can also have strange attractors. The depiction of such an attractor, in the case of a two-unit GRU, is provided in Figure 6 of the appendix.

## 2.2 CHAOS-FREE BEHAVIOR OF THE CFN

The dynamical behavior of the CFN is dramatically different from that of the LSTM. In this subsection we start by showing that the hidden states of the CFN activate and relax toward zero in a predictable fashion in response to input data. On one hand, this shows that the CFN cannot produce non-trivial dynamics without some influence from data. On the other, this leads to an interpretable model; any non-trivial activations in the hidden states of a CFN have a clear cause emanating from

data-driven activation. This follows from a precise, quantified description of the intuitive picture (3)–(4) sketched in the introduction.

We begin with the following simple estimate that sheds light on how the hidden states of the CFN activate and then relax toward the origin.

**Lemma 1.** *For any $T, k > 0$ we have*

$$|h_{T+k}(i)| \leq \Theta^k |h_T(i)| + \frac{H}{1 - \Theta} \left( \max_{T \leq t \leq T+k} |(Wx_t)(i)| \right)$$

*where $\Theta$ and $H$ are the maximum values of the $i^{\text{th}}$ components of the $\theta$ and $\eta$ gate in the time interval $[T, T + k]$, that is:*

$$\Theta = \max_{T \leq t \leq T+k} \theta_t(i) \quad and \quad H = \max_{T \leq t \leq T+k} \eta_t(i).$$

This estimate shows that if during a time interval $[T_1, T_2]$ one of

(i) the embedded inputs $Wx_t$ have weak $i^{\text{th}}$ feature (i.e. $\max_{T \leq t \leq T+k} |(Wx_t)(i)|$ is small),

(ii) or the input gates $\eta_t$ have their $i^{\text{th}}$ component close to zero (i.e. $H$ is small),

occurs then the $i^{\text{th}}$ component of the hidden state $h_t$ will relaxes toward zero at a rate that depends on the value of the $i^{\text{th}}$ component the the forget gate. Overall this leads to the following simple picture: $h_t(i)$ activates when presented with an embedded input $Wx_t$ with strong $i^{\text{th}}$ feature, and then relaxes toward zero until the data present the network once again with strong $i^{\text{th}}$ feature. The strength of the activation and the decay rate are controlled by the $i^{\text{th}}$ component of the input and forget gates. The proof of Lemma 1 is elementary —

*Proof of Lemma 1.* Using the non-expansivity of the hyperbolic tangent, i.e. $|\tanh(x)| \leq |x|$, and the triangle inequality, we obtain from (1)

$$|h_t(i)| \leq \Theta |h_{t-1}(i)| + H \max_{T \leq t \leq T+k} |(Wx_t)(i)|$$

whenever $t$ is in the interval $[T, T + k]$. Iterating this inequality and summing the geometric series then gives

$$|h_{T+k}(i)| \leq \Theta^k |h_T(i)| + \left( \frac{1 - \Theta^k}{1 - \Theta} \right) H \max_{T \leq t \leq T+k} |(Wx_t)(i)|$$

from which we easily conclude. $\square$

We now turn toward the analysis of the long-term behavior of the the dynamical system

$$\mathfrak{u}_t = h_t, \qquad \mathfrak{u} \mapsto \Phi(\mathfrak{u}) := \sigma \left( U_\theta \mathfrak{u} + b_\theta \right) \odot \tanh(\mathfrak{u}). \tag{10}$$

induced by a CFN. The following lemma shows that the only attractor of this dynamical system is the zero state.

**Lemma 2.** *Starting from any initial state $\mathfrak{u}_0$, the trajectory $\mathcal{O}^+(\mathfrak{u}_0)$ will eventually converge to the zero state. That is, $\lim_{t \to +\infty} \mathfrak{u}_t = 0$ regardless of the the initial state $\mathfrak{u}_0$.*

*Proof.* From the definition of $\Phi$ we clearly have that the sequence defined by $\mathfrak{u}_{t+1} = \Phi(\mathfrak{u}_t)$ satisfies $-1 < \mathfrak{u}_t(i) < 1$ for all $t$ and all $i$. Since the sequence $\mathfrak{u}_t$ is bounded, so is the sequence $\mathbf{v}_t := U_\theta \mathfrak{u}_t + b_\theta$. That is there exists a finite $C > 0$ such that $(U_\theta \mathfrak{u}_t)(i) + b_\theta(i) < C$ for all $t$ and $i$. Using the non-expansivity of the hyperbolic tangent, we then obtain that $|\mathfrak{u}_t(i)| \leq \sigma(C)|\mathfrak{u}_{t-1}(i)|$, for all $t$ and all $i$. We conclude by noting that $0 < \sigma(C) < 1$. $\square$

Lemma 2 remains true for a multi-layer CFN, that is, a CFN in which the first layer is defined by (1) and the subsequent layers $2 \leq \ell \leq L$ are defined by:

$$h_t^{(\ell)} = \theta_t^{(\ell)} \odot \tanh(h_{t-1}^{(\ell)}) + \eta_t^{(\ell)} \odot \tanh(W^{(\ell)} h_t^{(\ell-1)}).$$

Assume that $Wx_t = 0$ for all $t > T$, then an extension of the arguments contained in the proof of the two previous lemmas shows that

$$|h_{T+k}^{(\ell)}| \leq C(1 + k)^{(\ell-1)} \Theta^k \tag{11}$$

where $0 < \Theta < 1$ is the maximal values for the input gates involved in layer 1 to $\ell$ of the network, and $C > 0$ is some constant depending only on the norms $\|W^{(j)}\|_\infty$ of the matrices and the sizes $|h_T^{(j)}|$ of the initial conditions at all previous $1 \le j \le \ell$ levels. Estimate (11) shows that Lemma 2 remains true for multi-layer architectures.

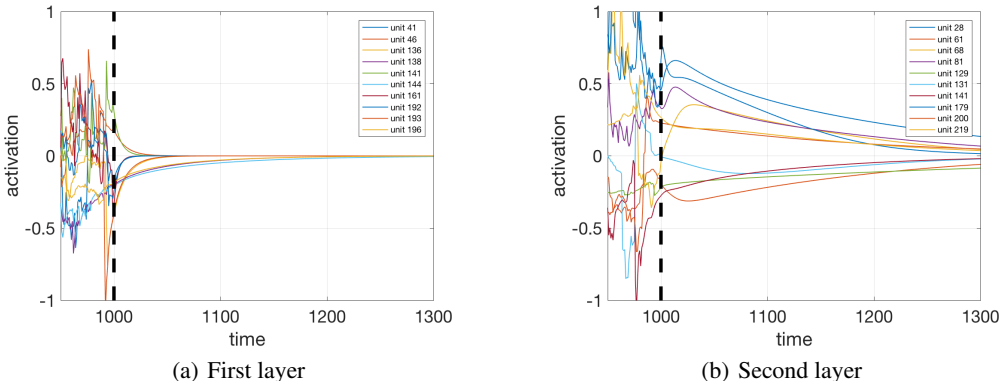

(a) First layer (b) Second layer

Figure 4: A 2-layer, 224-unit CFN trained on Penn Treebank. All inputs $x_t$ are zero after $t = 1000$, i.e. the time-point indicated by the dashed line. At left: plot of the 10 "slowest" units of the first layer. At right: plot of the 10 slowest units of the second layer. The second layer retains information much longer than the first layer.

Inequality (11) shows that higher levels (i.e. larger $\ell$) decay more slowly, and remain non-trivial, while earlier levels (i.e. smaller $\ell$) decay more quickly. We illustrate this behavior computationally with a simple experiment. We take a 2-layer, 224-unit CFN network trained on Penn Treebank and feed it the following input data: The first 1000 inputs $x_t$ are the first 1000 words of the test set of Penn Treebank; All subsequent inputs are zero. In other words, $x_t = 0$ if $t > 1000$. For each of the two layers we then select the 10 units that decay the slowest after $t > 1000$ and plot them on Figure 4. The figure illustrates that the second layer retains information for much longer than the first layer. To quantify this observation we define the relaxation time (or half-life) of the $i^{\text{th}}$ unit as the smallest $T$ such that

$$|h_{1000+T}(i)| < 0.5|h_{1000}(i)|.$$

Using this definition yields average relaxation times of 2.2 time steps for the first layer and 23.2 time steps for the second layer. The first layer has a standard deviations of approximately 5 steps while the second layer has a standard deviation of approximately 75 time steps. A more fine-grained analysis reveals that some units in the second layer have relaxation times of several hundred steps. For instance, if instead of averaging the relaxation times over the whole layer we average them over the top quartile (i.e. the 25% units that decay the most slowly) we get 4.8 time steps and 85.6 time steps for the first and second layers, respectively. In other words, by restricting attention to long-term units the difference between the first and second layers becomes much more striking.

Overall, this experiment conforms with the analysis (11), and indicates that adding a third or fourth layer would potentially allow a multi-layer CFN architecture to retain information for even longer.

## 3 EXPERIMENTS

In this section we show that despite its simplicity, the CFN network achieves performance comparable to the much more complex LSTM network on the word level language modeling task. We use two datasets for these experiments, namely the Penn Treebank corpus (Marcus et al., 1993) and the Text8 corpus (Mikolov et al., 2014). We consider both one-layer and two-layer CFNs and LSTMs for our experiments. We train both CFN and LSTM networks in a similar fashion and always compare models that use the same number of parameters. We compare their performance with and without dropout, and show that in both cases they obtain similar results. We also provide results published in Mikolov et al. (2014), Jozefowicz et al. (2015) and Sukhbaatar et al. (2015) for the sake of comparison.

Table 1: Experiments on Penn Treebank without dropout.

| Model | Size | Training | Val. perp. | Test perp. |
|---|---|---|---|---|
| Vanilla RNN | 5M parameters | Jozefowicz et al. (2015) | - | 122.9 |
| GRU | 5M parameters | Jozefowicz et al. (2015) | - | 108.2 |
| LSTM | 5M parameters | Jozefowicz et al. (2015) | - | 109.7 |
| LSTM (1 layer) | 5M parameters | Trained by us | 108.4 | 105.1 |
| CFN (2 layers) | 5M parameters | Trained by us | 109.3 | 106.3 |

Table 2: Experiments on Text8 without dropout

| Model | Size | Training | Perp. on development set |
|---|---|---|---|
| Vanilla RNN | 500 hidden units | Mikolov et al. (2014) | 184 |
| SCRN | 500 hidden units | Mikolov et al. (2014) | 161 |
| LSTM | 500 hidden units | Mikolov et al. (2014) | 156 |
| MemN2N | 500 hidden units | Sukhbaatar et al. (2015) | 147 |
| LSTM (2 layers) | 46.4M parameters | Trained by us | 139.9 |
| CFN (2 layers) | 46.4M parameters | Trained by us | 142.0 |

For concreteness, the exact implementation for the two-layer architecture of our model is

$$h_t^{(0)} = W^{(0)} x_t$$
$$\hat{h}_t^{(0)} = \mathrm{Drop}(h_t^{(0)}, p)$$
$$h_t^{(1)} = \theta_t^{(1)} \odot \tanh(h_{t-1}^{(1)}) + \eta_t^{(1)} \odot \tanh(W^{(1)} \hat{h}_t^{(0)})$$
$$\hat{h}_t^{(1)} = \mathrm{Drop}(h_t^{(1)}, p)$$
$$h_t^{(2)} = \theta_t^{(2)} \odot \tanh(h_{t-1}^{(2)}) + \eta_t^{(2)} \odot \tanh(W^{(2)} \hat{h}_t^{(1)})$$
$$\hat{h}_t^{(2)} = \mathrm{Drop}(h_t^{(2)}, p)$$
$$y_t = \mathrm{LogSoftmax}(W^{(3)} \hat{h}_t^{(2)} + b)$$

where $\mathrm{Drop}(z, p)$ denotes the dropout operator with a probability $p$ of setting components in $z$ to zero. We compute the gates according to

$$\theta_t^{(\ell)} := \sigma\left( U_\theta^{(\ell)} \tilde{h}_{t-1}^{(\ell)} + V_\theta^{(\ell)} \tilde{h}_t^{(\ell-1)} + b_\theta \right) \quad \text{and} \quad \eta_t^{(\ell)} := \sigma\left( U_\eta^{(\ell)} \tilde{h}_{t-1}^{(\ell)} + V_\eta^{(\ell)} \tilde{h}_t^{(\ell-1)} + b_\eta \right)$$

where $\quad \tilde{h}_{t-1}^{(\ell)} = \mathrm{Drop}(h_{t-1}^{(\ell)}, q) \quad$ and $\quad \tilde{h}_t^{(\ell-1)} = \mathrm{Drop}(h_t^{(\ell-1)}, q),$

and thus the model has two dropout hyperparameters. The parameter $p$ controls the amount of dropout between layers; the parameter $q$ controls the amount of dropout inside each gate. We use a similar dropout strategy for the LSTM, in that all sigmoid gates $f, o$ and $i$ receive the same amount $q$ of dropout.

To train the CFN and LSTM networks, we use a simple online steepest descent algorithm. We update the weights $w$ via

$$w^{(k+1)} = w^{(k)} - \mathrm{lr} \cdot \vec{p} \quad \text{where} \quad \vec{p} = \frac{\nabla_w L}{\|\nabla_w L\|_2}, \tag{12}$$

where lr is the learning rate and $\nabla_w L$ denotes the approximate gradient of the loss with respect to the weights as estimated from a certain number of presented examples. We use the usual backpropagation through time approximation when estimating the gradient: we unroll the net $T$ steps in the past and neglect longer dependencies. In all experiments, the CFN and LSTM networks are unrolled for $T = 35$ steps and we take minibatches of size 20. As all search directions $\vec{p}$ have Euclidean norm $\|\vec{p}\|_2 = 1$, we perform no gradient clipping during training.

We initialize all the weights in the CFN, except for the bias of the gates, uniformly at random in $[-0.07, 0.07]$. We initialize the bias $b_\theta$ and $b_\eta$ of the gates to 1 and $-1$, respectively, so that at the beginning of the training $\theta_t \approx \sigma(1) \approx 0.73$ and $\eta_t \approx \sigma(-1) \approx 0.23$. We initialize the weights of the LSTM in exactly the same way; the bias for the forget and input gate are initialized to 1 and $-1$, and all the other weights are initialized uniformly in $[-0.07, 0.07]$. This initialization scheme favors

Table 3: Experiments on Penn Treebank with dropout.

| Model | Size | Training | Val. perp. | Test perp. |
|---|---|---|---|---|
| Vanilla RNN | 20M parameters | Jozefowicz et al. (2015) | 103.0 | 97.7 |
| GRU | 20M parameters | Jozefowicz et al. (2015) | 95.5 | 91.7 |
| LSTM | 20M parameters | Jozefowicz et al. (2015) | 83.3 | 78.8 |
| LSTM (2 layers) | 20M parameters | Trained by us | 78.4 | 74.3 |
| CFN (2 layers) | 20M parameters | Trained by us | 79.7 | 74.9 |
| LSTM (2 layers) | 50M parameters | Trained by us | 75.9 | 71.8 |
| CFN (2 layers) | 50M parameters | Trained by us | 77.0 | 72.2 |

the flow of information in the horizontal direction. The importance of a careful initialization of the forget gate was pointed out in Gers et al. (2000) and Jozefowicz et al. (2015). Finally, we initialize all hidden states to zero for both models.

**Dataset Construction.** The Penn Treebank Corpus has 1 million words and a vocabulary size of 10,000. We used the code from Zaremba et al. (2014) to construct and split the dataset into a training set (929K words), a validation set (73K words) and a test set (82K words). The Text8 corpus has 100 million characters and a vocabulary size of 44,000. We used the script from Mikolov et al. (2014) to construct and split the dataset into a training set (first 99M characters) and a development set (last 1M characters).

**Experiments without Dropout.** Tables 1 and 2 provide a comparison of various recurrent network architectures without dropout evaluated on the Penn Treebank corpus and the Text8 corpus. The last two rows of each table provide results for LSTM and CFN networks trained and initialized in the manner described above. We have tried both one and two layer architectures, and reported only the best result. The learning rate schedules used for each network are described in the appendix.

We also report results published in Jozefowicz et al. (2015) were a vanilla RNN, a GRU and an LSTM network were trained on Penn Treebank, each of them having 5 million parameters (only the test perplexity was reported). Finally we report results published in Mikolov et al. (2014) and Sukhbaatar et al. (2015) where various networks are trained on Text8. Of these four networks, only the LSTM network from Mikolov et al. (2014) has the same number of parameters than the CFN and LSTM networks we trained (46.4M parameters). The vanilla RNN, Structurally Constrained Recurrent Network (SCRN) and End-To-End Memory Network (MemN2N) all have 500 units, but less than 46.4M parameters. We nonetheless indicate their performance in Table 2 to provide some context.

**Experiments with Dropout.** Table 3 provides a comparison of various recurrent network architectures with dropout evaluated on the Penn Treebank corpus. The first three rows report results published in (Jozefowicz et al., 2015) and the last four rows provide results for LSTM and CFN networks trained and initialized with the strategy previously described. The dropout rate $p$ and $q$ are chosen as follows: For the experiments with 20M parameters, we set $p = 55\%$ and $q = 45\%$ for the CFN and $p = 60\%$ and $q = 40\%$ for the LSTM; For the experiments with 50M parameters, we set $p = 65\%$ and $q = 55\%$ for the CFN and $p = 70\%$ and $q = 50\%$ for the LSTM.

## 4 CONCLUSION

Despite its simple dynamics, the CFN obtains results that compare well against LSTM networks and GRUs on word-level language modeling. This indicates that it might be possible, in general, to build RNNs that perform well while avoiding the intricate, uninterpretable and potentially chaotic dynamics that can occur in LSTMs and GRUs. Of course, it remains to be seen if dynamically simple RNNs such as the proposed CFN can perform well on a wide variety of tasks, potentially requiring longer term dependencies than the one needed for word level language modeling. The experiments presented in Section 2 indicate a plausible path forward — activations in the higher layers of a multi-layer CFN decay at a slower rate than the activations in the lower layers. In theory, complexity and long-term dependencies can therefore be captured using a more "feed-forward" approach (i.e. stacking layers) rather than relying on the intricate and hard to interpret dynamics of an LSTM or a GRU.

Overall, the CFN is a simple model and it therefore has the potential of being mathematically well-understood. In particular, Section 2 reveals that the dynamics of its hidden states are inherently more interpretable than those of an LSTM. The mathematical analysis here provides a few key insights into the network, in both the presence and absence of input data, but obviously more work is needed before a complete picture can emerge. We hope that this investigation opens up new avenues of inquiry, and that such an understanding will drive subsequent improvements.

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

APPENDIX

**Strange attractor of the Hénon map.** For the sake of comparison, we provide in Figure 5 a depiction of a well-known strange attractor (the Hénon attractor) arising from a discrete-time dynamical system. We generate these pictures by reproducing the numerical experiments from Hénon (1976). The discrete dynamical system considered here is the two dimensional map

$$x_{t+1} = y_t + 1 - ax_t^2, \quad y_{t+1} = bx_t,$$

with parameters set to $a = 1.4$ and $b = 0.3$. We obtain Figure 5(a) by choosing the initial state $(x_0, y_0) = (0, 0)$ and plotting the iterates $(x_t, y_t)$ for $t$ between $10^3$ and $10^5$. All trajectories starting close to the origin at time $t = 0$ converge toward the depicted attractor. Successive zooms on the branch of the attractor reveal its fractal nature. The structure repeats in a fashion remarkably similar to the 2-unit LSTM in Section 2.

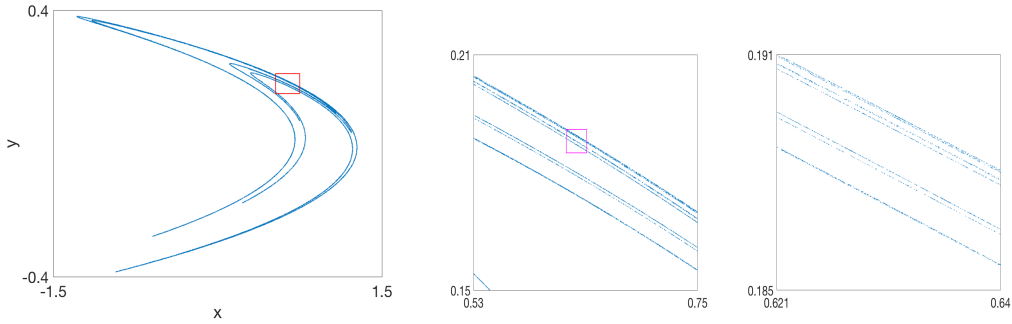

Figure 5: Strange attractor of the Hénon map. From left to right: The Hénon attractor, enlargement of the red box, enlargement of the magenta box.

**Strange attractor of a 2-unit GRU.** As with LSTMs, the GRU gated architecture can induce a chaotic dynamical system. Figure 6 depicts the strange attractor of the dynamical system

$$\mathfrak{u}_t = h_t, \qquad \mathfrak{u} \mapsto \Phi(\mathfrak{u}) := (1 - z) \odot \mathfrak{u} + z \odot \tanh\left(U(r \odot \mathfrak{u})\right)$$
$$z := \sigma\left(W_z\mathfrak{u} + b_z\right) \quad r := \sigma\left(W_r\mathfrak{u} + b_r\right),$$

induced by a two-dimensional GRU, with weight matrices

$$W_z = \begin{bmatrix} 0 & 1 \\ 1 & 1 \end{bmatrix} \quad W_r = \begin{bmatrix} 0 & 1 \\ 1 & 0 \end{bmatrix} \quad U = \begin{bmatrix} -5 & -8 \\ 8 & 5 \end{bmatrix}$$

and zero bias for the model parameters. Here also successive zooms on the branch of the attractor reveal its fractal nature. As in the LSTM, the forward trajectories of this dynamical system exhibit a high degree of sensitivity to initial states.

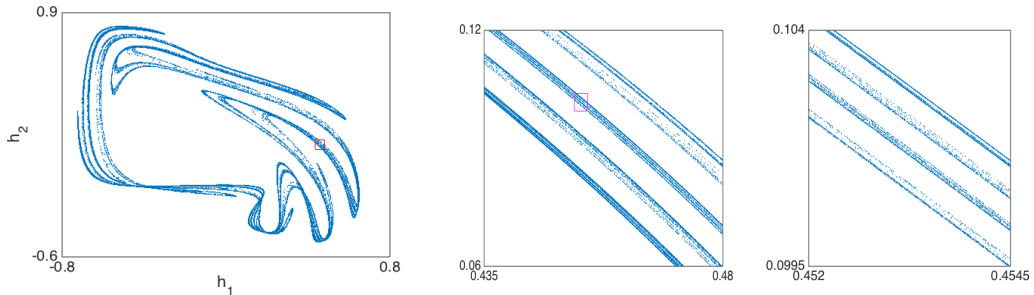

Figure 6: Strange attractor of a two-unit GRU. Successive zooms reveal the fractal nature of the attractor.

**Network sizes and learning rate schedules used in the experiments.** In the Penn Treebank experiment without dropout (Table 1), the CFN network has two hidden layers of 224 units each for a total of 5 million parameters. The LSTM has one hidden layer with 228 units for a total of 5 million parameters as well. We also tried a two-layer LSTM with 5 million parameters but the result was worse (test perplexity of 110.6) and we did not report it in the table. For the Text8 experiments (Table 2), the LSTM has two hidden layers with 481 hidden units for a total 46.4 million parameters. We also tried a one-layer LSTM with 46.4 million parameters but the result was worse (perplexity of 140.8). The CFN has two hidden layers with 495 units each, for a total of 46.4 million parameters as well.

For both experiments without dropout (Table 1 and 2), we used a simple and aggressive learning rate schedule: at each epoch, lr is divided by 3. For the CFN the initial learning rate was chosen to be $lr_0 = 5.5$ for PTB and $lr_0 = 5$ for Text8. For the LSTM we chose $lr_0 = 7$ for PTB and $lr_0 = 5$ for Text8.

In the Penn Treebank experiment with dropout (Table 3), the CFN with 20M parameters has two hidden layers of 731 units each and the LSTM with 20M parameters trained by us has two hidden layers of 655 units each. We also tried a one-layer LSTM with 20M parameters and it led to similar but slightly worse results than the two-layer architecture. For both network, the learning rate was divided by 1.1 each time the validation perplexity did not decrease by at least $1\%$. The initial learning rate were chosen to be $lr_0 = 7$ for the CFN and $lr_0 = 5$ for the LSTM.

