# Peer review of "A recurrent neural network without chaos"

_ICLR 2017 — accepted_

[Public Comment · Heikki Arponen · 07 Nov 2016]
**Edge of chaos?**

I think it would be useful to discuss the concept of *edge* of chaos here (see e.g. Bertschinger, Nachschlager - Real-Time Computation at the Edge of Chaos in Recurrent Neural Networks), i.e. the hypothesis that RNNs are optimal (in some sense) at the boundary between chaotic and deterministic regimes. Specifically, it would be nice to see if your network gets closer to this edge during training (I think it will).

It wasn't clear to me if you studied the chaoticity in the case *with* input... the "epsilon-activation" thing seems very nonstandard. Why didn't you just compute the mean Lyapunov exponent? You can do that with or without input. I think you might find that the RNN with input will approach the edge of chaos during training (Lyapunov exp gets closer to zero, probably starting from negative values in your case).

The LSTM phase space diagram in Fig. 2 looks pretty bad... I think that particular unit is not behaving well at all. What you should get in properly trained models is something like in Fig. 1 (a), but more noisy because there's effects from the input.

Anyway, overall a very interesting paper! I'm glad to see RNNs studied from a chaotic dynamics perspective.

[Author Response · Thomas Laurent · 12 Dec 2016]
**Authors’ comment: Added experiments on long term dependencies.**

Several reviewers have posted comments asking about the capability of the proposed model to capture long-term dependencies. This is a natural question since the model was designed so that units get activated when presented the correct feature, then relax to zero at a rate controlled by the forget gate. At a first glance it is unclear that such a simple mechanism could capture long term dependencies (the relaxing rates might be too fast).

We added a simple experiment in the paper showing that long term dependencies can be obtained by stacking multiple layers of the basic architecture (see Figure 4). We took a 2-layer, 224-unit CFN network trained on Penn Treebank and ran it with the following input data: The first 1000 inputs x_t are the first 1000 words of the test set of PTB; All subsequent inputs are set to zero, so that x_t=0 if t>1000. For each layer we then select the 10 units that decay the slowest after t>1000 and plotted them on Figure 4. The first layer retains information for about 10 time steps, whereas the second layer retains information for about 100 steps. Adding a third or fourth layer would then allow the architecture to retain information for even longer periods. We have not yet implemented a multi-layer network to handle tasks (other than language modeling) where such longer-term dependencies are needed, but we believe the main obstacle here is one of proper initialization and training rather than a shortcoming of the architecture itself.

Importantly, this behavior (i.e. higher layers decay more slowly) can be explained analytically, see equation (11).

Overall, we find it interesting that complexity and long-term dependencies can plausibly be obtained in a classical way (i.e. stacking layers) rather than relying on the intricate and hard to interpret dynamics of an LSTM.

[Author Response · Thomas Laurent · 16 Dec 2016]
**Authors’ comment: Conclusion added**

We added a short conclusion reflecting some of the discussions with the reviewers.

[Official Review · AnonReviewer2 · rating 7 · confidence 4 · 17 Dec 2016 (modified: 18 Dec 2016)]
**Cool paper**
soundness 5 · originality 4

This paper poses an interesting idea: removing chaotic behavior or RNNs.
While many other papers on new RNN architecture usually focus too much on the performance improvement and leave the analysis part on their success as a black-box, this paper does a good job on presenting why its method may work well.

Although, the paper shows lots of comparison between the chaotic systems (GRUs & LSTMs) and the stable system (proposed CFN model), the reviewer is not fully convinced by the main claim of this paper, the nuance that chaotic behaviour makes dynamic system to have rich representation power but makes the system too unstable. In the paper, the LSTM shows a very sensitive behaviour, even when a very small amount of noise is added to the input. However, it still performs surprisingly well with this chaotic behaviour. 

Measuring the model complexity is a very difficult task, therefore, many papers manage to use either same number of hidden units or choose approximately close model sizes. In this paper, the experiments were carried by using the same amount of parameters for both the LSTM and CFN. However, I think the CFN may have much more simpler computational graph. Taking the idea of this work, can we develop a stable dynamic system, but which does not only have one attractor?

It is also interesting to see that the layers of CFNs are updated in different timescales in a sense that the decaying speed decreases when the layer gets higher. Could you provide more statistics on this? For example, what is the average relaxation time of the whole hidden units at each layer?

Batch normalization and layer normalization can be helpful to make the training of RNNs become more stable. How would the behaviour of batch normalized or perhaps layer normalized LSTM look like? Also, it is often not trivial to make batch normalization or layer normalization to work on a new architecture. I think it may be useful to compare batch normalized or layer normalized versions of the LSTM and CFN.

The quality of the work is good, explanation is clear enough along with nice analyses and proofs. Overall, the performance is not any better than LSTMs, but it is still interesting when thinking of simplicity of this model. I am a bit concerned if this model might not work that well in more harder task, e.g., translation. Figure 4 of this paper is very interesting, where the proposed architecture shows that the hidden units at the second layer tends to keep its information longer than the first layer ones.

[Official Review · AnonReviewer3 · rating 7 · confidence 4 · 18 Dec 2016]
**interesting starting point**
soundness 4 · originality 3 · impact 4

I think the authors provide an interesting direction for understanding and maybe constructing recurrent models that are easier to interpret. Is not clear where such direction will lead but I think it could be an interesting starting point for future work, one that worth exploring.

[Official Review · AnonReviewer1 · rating 8 · confidence 3 · 20 Dec 2016]
**Nice investigation**
originality 5

The authors of the paper set out to answer the question whether chaotic behaviour is a necessary ingredient for RNNs to perform well on some tasks. For that question's sake, they propose an architecture which is designed to not have chaos. The subsequent experiments validate the claim that chaos is not necessary.

This paper is refreshing. Instead of proposing another incremental improvement, the authors start out with a clear hypothesis and test it. This might set the base for future design principles of RNNs.

The only downside is that the experiments are only conducted on tasks which are known to be not that demanding from a dynamical systems perspective; it would have been nice if the authors had traversed the set of data sets more to find data where chaos is actually necessary.

[Public Comment · Greg Yang · 31 Dec 2016]
**Questions**

Thanks for a very interesting read.

What happens if instead of driving the LSTMs with x_t = 0, you drive it with a fixed input, like the word "What"? Would that behave the same as in fig 3?

If you drive the LSTMs with some input and then fix x_t = 0 for t > T (as in fig 4), do you still see chaos? If there is gradual decay in the hidden units' activations, do you also see that the second layer forgets more slowly than the first?

Have you tried training on the copy task as in the algorithmic learning literature (like NTM), to see whether there is a actual difference in how long memory is retained in CFN vs LSTM?

[Final Decision · Program Chairs · 06 Feb 2017]
**ICLR committee final decision**

The reviewers all enjoyed this paper and the analysis.
 
 pros:
 - novel new model
 - interesting insights into the design of model, through analysis of trajectories of hidden states of RNNs.
 
 cons:
 - results are worse than LSTMs.